

# Effects of 4-week velocity-based HIIT on athletic performance in youth soccer players

Murat Koç[1], Niyazi Sıdkı Adıgüzel[2], Hakan Engin[3], Barışcan Öztürk[3], Umut Canli[4], Aydın Karaçam[2], Bekir Erhan Orhan[5], Pablo Prieto-González[6], Peter Bartik[6], Shahad Alghemlas[6], Maria Isip[6] and Peter Sagat[6]

[1] Faculty of Sport Sciences, Erciyes University, Kayseri, Turkey
[2] Faculty of Sport Sciences, Bandırma Onyedi Eylül University, Balıkesir, Turkey
[3] Faculty of Sport Sciences, Çukurova University, Adana, Turkey
[4] Faculty of Sport Sciences, Tekirdağ Namık Kemal University, Tekirdağ, Turkey
[5] Faculty of Sport Sciences, İstanbul Aydın University, İstanbul, Turkey
[6] Sport Sciences and Diagnostics Research Group, College of Humanities and Sciences, Prince Sultan University, Riyadh, Saudi Arabia

Corresponding authors
Pablo Prieto-González,
pprieto@psu.edu.sa
Peter Sagat, sagat@psu.edu.sa

## ABSTRACT

**Objective:** Young soccer players need to enhance their athletic performance, including speed and endurance. Traditional training methods may not be effective enough to improve athletic performance in these young athletes. Velocity-based high-intensity interval training (vHIIT) workouts can increase the efficiency of energy systems and improve athletic performance. This study aimed to investigate the effects of four weeks of vHIIT on athletic performance in young soccer players.

**Method:** A total of 14 male soccer players participated in the study (mean age: 18.9 ± 1.0 years, body mass: 76.5 ± 5.3 kg, height: 1.81 ± 0.08 m). Participants were randomly assigned to either an experimental group ($n = 7$) or a control group ($n = 7$). While the control group continued their regular soccer training without additional vHIIT intervention, the experimental group underwent vHIIT training at 85–90% intensity twice a week for 4 weeks in addition to their regular training. Change of direction speed (COD), maximum sprint speed (MSS), maximum oxygen consumption ($VO_{2max}$), and the 30-15 intermittent fitness test (VIFT) were assessed twice, in the control and experimental groups, both at pre-test and post-test measurements.

**Results:** The findings indicated a significant decrease in COD time ($p < 0.001$, $\eta^2 p = 0.682$) and 30-15 IFT ($p < 0.001$, $\eta^2 p = 0.735$) in the experimental group. However, no statistically significant change was observed in these parameters between the pre-test and post-test in the control group. Additionally, group-time interaction effects were found to be significant in favor of the experimental group in all parameters.

**Conclusion:** Four weeks of speed-based vHIIT training led to improvements in sprint performance, COD, and aerobic capacity in young soccer players. Incorporating vHIIT workouts into conditioning programs for youth soccer players may be an effective strategy for enhancing physical performance components, including speed, agility, and endurance, which are requirements of soccer.

# INTRODUCTION

Soccer is an intermittent sport that requires a combination of physical, technical, and cognitive abilities, depending on position and competition level (*Joksimović et al., 2019*). Success in this sport demands a highly developed combination of anaerobic and aerobic fitness characteristics (*Slimani et al., 2019*). It is estimated that approximately 90% of the total energy cost during a soccer match is provided by aerobic metabolism (*Cihan, 2022*; *Radziminski et al., 2013*). Players typically cover a distance of 8 to 12 km during a match, and aerobic capacity plays a crucial role in recovery between high-intensity sprints (*Helgerud et al., 2011*). The ability to sustain repeated high-intensity efforts is critical for offensive and defensive strategies, making it a fundamental component of team success (*Michalczyk et al., 2010*). In a professional soccer match, the high-intensity running distance accounts for approximately 6% to 13.5% of the total distance covered (*Dellal et al., 2011*). High-intensity running, a key physical activity indicator in elite soccer, is directly related to players' decisive, short, and intense actions (*Djaoui et al., 2014*). To meet these demands, high-intensity interval training (HIIT) emerges as an effective method for preparing for the high-intensity physical efforts required in soccer.

HIIT training is crucial to athletic conditioning, particularly for sports that require sustained high-intensity efforts. This training method typically consists of short bursts of intense exercise followed by periods of rest or low-intensity activity (*Bisciotti et al., 2020*). Research has shown that HIIT positively impacts aerobic and anaerobic capacities, significantly improving $VO_{2max}$ and anaerobic threshold, which are crucial for soccer players (*Jatmiko, Kusnanik & Sidik, 2024*). The training method of HIIT controls the workload recovery balance by tracking athlete-specific minute heart rate (HR) responses (*Wong et al., 2010*). The physiological signal of heart rate is an essential exercise parameter for adjusting training intensity and recovery periods (*Choi & Roh, 2018*; *Jones, Griffiths & Mellalieu, 2017*). This approach does not directly align with the maximum distance achieved in pre-established aerobic endurance milestones (*Makni et al., 2023*; *Tanisho & Hirakawa, 2009*). The utilization of HR as a commonly adopted approach for HIIT intensity prescription faces challenges because it relies on athletes' mental state and ability to control running intensity. The association between HR functioning and metabolic requirements remains insignificant (*Buchheit & Laursen, 2013*). The application of HR-based HIIT is effective for endurance sports, but speed-based HIIT yields better results when sprinting speed is a key factor in athletic competitions (*Arazi et al., 2017*; *Eddolls et al., 2017*; *Paquette et al., 2017*; *Reljic et al., 2019*; *Sal-de-Rellan et al., 2024*). The training method known as vHIIT improves sprint abilities when applied to subjects. vHIIT involves progressive intermittent shuttle-based field training and is designed using the speed results obtained from the 30-15 Intermittent Fitness Test (30-15 VIFT). vHIIT-designed training is more accurate and practical than HR-based HIIT for standardizing training intensity at the team level, as it brings players with different physiological profiles to a similar

cardiovascular demand level (*Arazi et al., 2017*). In vHIIT training, high-intensity workloads develop an athlete's anaerobic capacity while simultaneously challenging aerobic capacity, thereby improving endurance levels. Consequently, while enhancing anaerobic capacity, high-intensity training also facilitates the adaptations for athletes to sustain their $VO_{2max}$ speed for longer durations (*Laursen & Buchheit, 2019*).

The physical and physiological effects of HIIT training, designed based on HR in young soccer players, have been examined in several studies (*Fernandez-Galvan et al., 2022*; *Gabryś et al., 2019*; *Rabbani et al., 2019*). In a 6-week combined study of small-sided game (SSG) and HIIT, VIFT scores were found to increase from an average of 19.55 to 20.75 km/h, or 6.2%, and in the HIIT+SSG group, it increased from an average of 19.18 to 20.50 km/h, or 6.9% (*Rabbani et al., 2019*). A study investigating the effects of 4-week HIIT training observed that the maximum sprint speed of athletes with an average age of 17.4 increased from 17.05 to 17.30 km/h. (*Faude et al., 2013*). It has been reported that $VO_{2max}$ in male soccer players ranges from 50 to 75 ml/kg/min (*Stølen et al., 2005*). A review found that $VO_{2max}$ ranges more widely in young athletes. Additionally, in the HIIT groups, the average percentage increase from pre- to post-$VO_{2max}$ was found to be $7.2 \pm 6.9\%$ (*Engel et al., 2018*).

However, research investigating the effects of vHIIT training on athletic performance parameters —such as maximum sprint speed, change of direction speed (COD), 30-15 VIFT, and maximum oxygen consumption ($VO_{2max}$)—is limited (*Arazi et al., 2017*). Additionally, young soccer players exhibit different physiological adaptation mechanisms (such as anaerobic energy systems and recovery processes) than adult soccer players (*Calleja-Gonzalez et al., 2021*; *Radziminski et al., 2013*; *Sellami et al., 2024*). Therefore, training programs for young soccer players should be structured to consider individual adaptation processes. Unlike traditional HR-based HIIT, vHIIT training is adapted according to individual speed capacity, making the training load more personalized and enhancing the development of both anaerobic and aerobic capacities more effectively. While the contributions of HIIT to physical performance are well-documented in the literature, studies investigating the effectiveness of vHIIT training specifically for soccer players remain limited. The present study aims to evaluate the effects of a vHIIT training program on field-based physical and physiological performance parameters closely related to soccer matches in young soccer players. The study aims to provide scientific evidence for optimizing soccer training programs by examining the impact of vHIIT training on maximum sprint speed, change of direction speed (COD), 30-15 VIFT, and $VO_{2max}$ in young soccer players.

## MATERIALS AND METHODS

### Research design and participants

A randomized controlled trial was conducted in this study to determine the effects of the 4-week vHIIT training program on maximum sprint speed, change of direction speed (COD), the 30-15 VIFT, and $VO_{2max}$ (Fig. 1). The study was approved by the Ethics Committee of Bandırma Onyedi Eylül University, Health Sciences Non-Interventional Research (20-12-2024-1042/241).

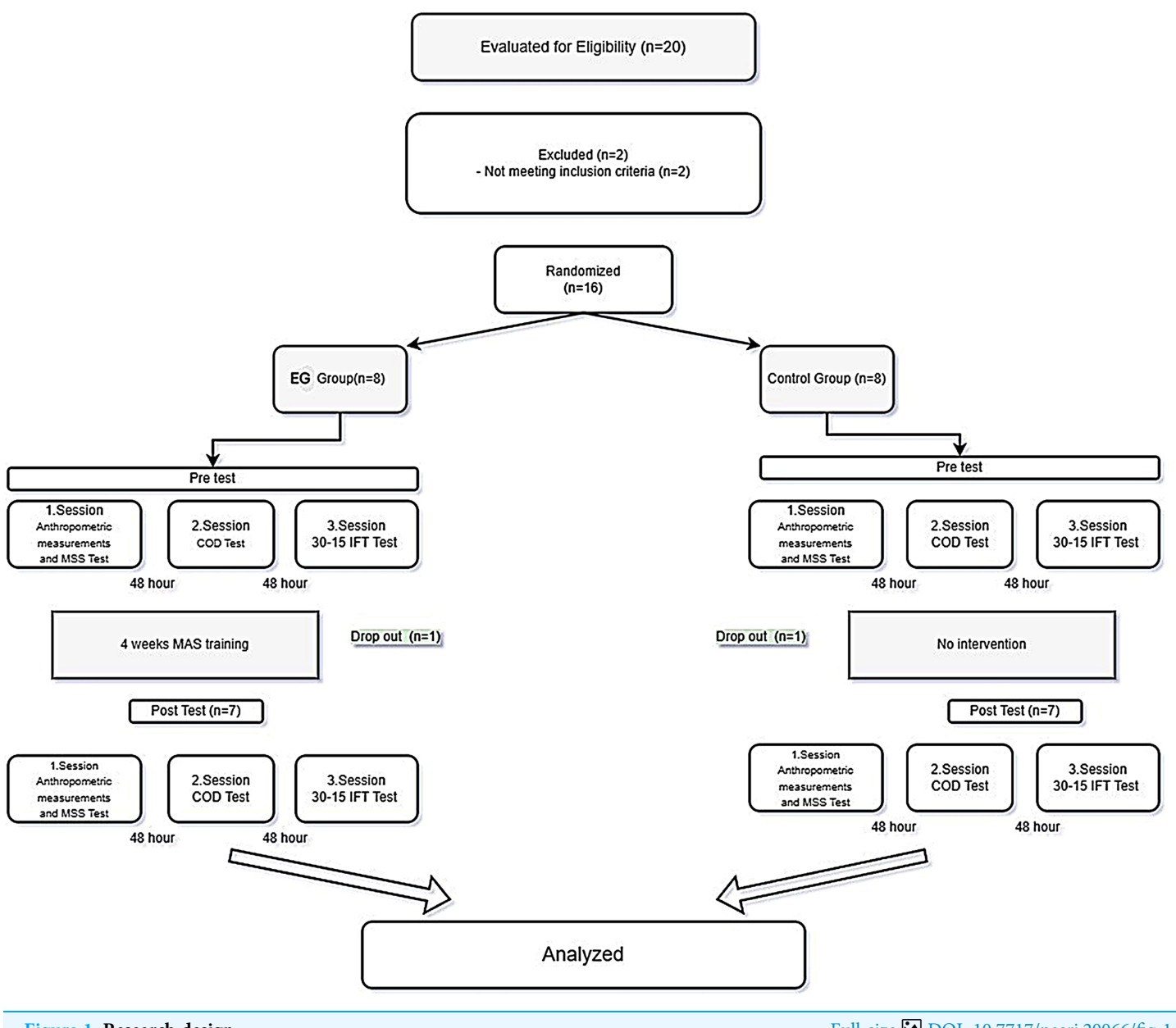

**Figure 1 Research design.**

## Determination of the number of participants

To ascertain the necessary sample size for this study, G*Power (version 3.1) was utilized to perform a power analysis. This analysis employed an F-test for ANOVA with repeated measures and an interaction design. The power analysis incorporated input parameters such as an effect size of f = 0.25 (*Thomas et al., 2022*), an alpha error probability ($\alpha$) of 0.05, and a statistical power ($1-\beta$) of 0.80. With two groups involved, the analysis determined that a minimum of 34 participants was necessary to achieve statistical significance (*Faul et al., 2009*).

**Table 1 Demographic characteristics of the participants.**

| Group | Age (year) $\bar{x} \pm SS$ | Height (m) $\bar{x} \pm SS$ | Body mass (kg) $\bar{x} \pm SS$ | BMI $\bar{x} \pm SS$ | Sports age (year) $\bar{x} \pm SS$ |
|---|---|---|---|---|---|
| Exp (n = 7) | 19.14 ± 0.90 | 1.82 ± 0.10 | 75.86 ± 6.49 | 22.80 ± 1.09 | 8.00 ± 1.00 |
| Con (n = 7) | 18.71 ± 1.11 | 1.81 ± 0.06 | 77.14 ± 4.30 | 23.51 ± 0.49 | 9.00 ± 1.15 |

**Note:**
Exp, experimental; Con, control.

## Participants

This study was completed with 14 male soccer players playing in a semi-professional team. All participants were from the same club. Written consent forms were obtained from all participants. Participants were assigned to experimental (vHIIT) and control groups using simple random assignments. Of the 20 athletes who initially participated in the study voluntarily, four dropped out due to injury, and two were excluded because of incomplete data (missing test results) (Table 1; Fig. 1). As a result, a total of 14 athletes (age: 18.9 ± 1.0 years; body mass: 76.5 ± 5.3 kg; height: 1.81 ± 0.08 m) were included in the study. The participants had not previously participated in any additional endurance training and had not suffered any injuries in the last 3 months.

## Experimental design

vHIIT performed at 85–90% intensities and has significantly improved performance indicators, including power output and fatigue resistance (Arazi et al., 2017). Therefore, in addition to their regular soccer training, the Exp group underwent vHIIT training twice a week for four weeks at an intensity of 85–90% VIFT. These vHIIT sessions were performed in addition to the 4–5 weekly team sessions. The vHIIT levels were determined based on the 30-15 IFT results conducted during the pre-test. Each training session included three sets of seven repetitions, with each repetition consisting of 20 s of exertion followed by a rest period of 10 to 20 s. To ensure optimal recovery, 3-min rest periods were provided between sets. Participants in the control group continued with their regular soccer training only. Soccer players in the control group continued their regular soccer training in the same experienced classes 4–5 times a week (consisting of technical drills, tactical games and small-sided games lasting approximately 90 min). The applied training protocol is illustrated in Fig. 2.

## Procedures

This study examined the effects of the vHIIT training program on soccer players' athletic performance. Participants were assigned to the training (vHITT) and control groups using a simple random assignment method. To minimize assignment bias, care was taken to apply the same procedure to all participants. The training program was implemented based on the load progression principle, with training intensity increasing every 2 weeks (Knapik et al., 2012). Training loads were adjusted based on individual speed capacities, and workload and recovery durations were monitored. Specifically, players followed an individualized protocol in which heart rate measurements were used to monitor recovery

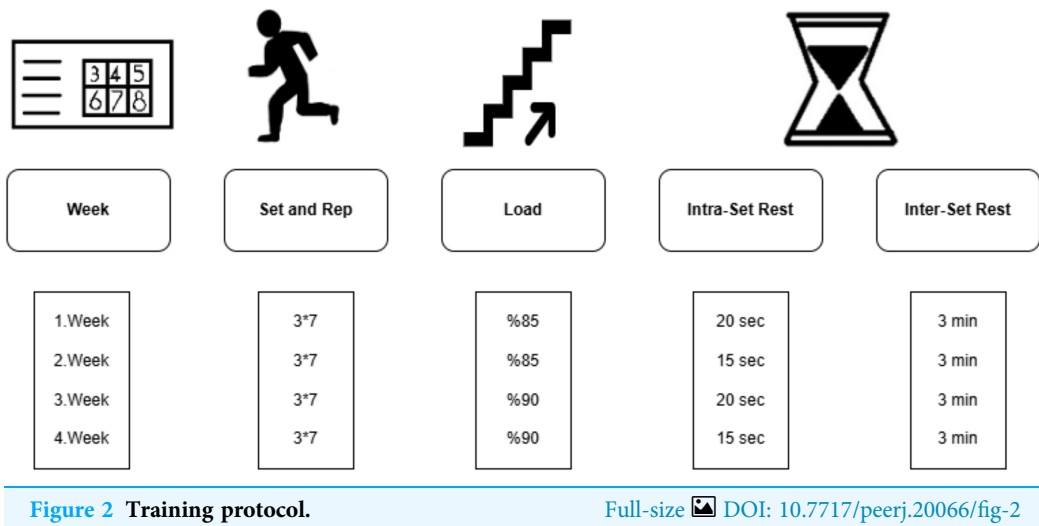

**Figure 2 Training protocol.**

processes after exertion. An athletic performance coach supervised all training sessions, with each coach overseeing five soccer players. Pre-tests were conducted in three separate sessions at 48-h intervals. In the first session, anthropometric measurements and a maximum sprint speed test were conducted. The second session included the change of direction speed (COD) test, while the final session involved the 30-15 IFT test. All tests were conducted under appropriate environmental conditions at the semi-professional Adana 5 Ocak Sports Club facilities. Sprint, COD, and 30-15 IFT tests were performed on an internationally standardized grass field, whereas anthropometric measurements were taken in the club's athletic performance studio at sea level, under controlled temperature and humidity conditions (average temperature: 24 °C, average humidity: 29%, barometric pressure range: 1,010–1,025 mmHg). To minimize the effects of circadian rhythm, the same researcher conducted pre-tests and post-tests simultaneously (17:00–18:00). Each test session included a 15-min general warm-up protocol and a 4-min passive rest period before applying group-specific protocols. Since the experimental and control groups were in the same club, training programs were scheduled at different times to prevent a possible "cross-contamination" risk. The vHIIT protocol was applied only to the experimental group at a certain time before the training. Additionally, during each exercise repetition and at moments when maintaining an intensity level was crucial, verbal encouragement was provided using predetermined standard phrases by the researcher (*e.g.*, "keep going," "you're doing great," "hold on a little longer"). This encouragement was not provided only when a decline in the participant's performance was observed, but was given throughout each trial with similar frequency and content to ensure an equal motivational environment for all participants. In this way, the effect of extrinsic motivation was attempted to be minimized (*Andreacci et al., 2002*).

## Data collection tools

Pre- and post-intervention assessments were conducted by the same experienced researcher. Standard protocols were strictly followed to minimise measurement bias.

## Body mass and height measurement

The athletes' height (with 0.5 cm sensitivity) and body mass (with 100 g sensitivity) were measured using a digital Seca stadiometer with a height scale. The demographic characteristics of the participants (age and training age) were determined using a questionnaire prepared by the researchers (*Adıgüzel et al., 2024*).

## Maximal sprint speed

The participants' maximal speeds were measured using a 10-m sprint test. Soccer players ran at maximum speed on a 40-m track, and their transition times over the last 10 m (between the 30th and 40th m) were recorded using photocells. Their hourly speed was calculated based on the distance covered and recorded time (*Al Haddad et al., 2015*; *Öztürk et al., 2023*). This test was conducted three times, and the best performance of each player was recorded.

## Change of direction test

For COD, a zigzag test was performed using the Witty Microgate photocell system (Version 1.6, Bolzano, Italy). The test consisted of three slaloms run 5 m apart, in a zigzag formation at $100°$ angles to each other, covering a distance of 20 m. Athletes performed the three slaloms at their highest speed, starting 1 m behind the starting line. It was performed three times for each athlete, and the best times were recorded (*Adıgüzel et al., 2024*; *Chaouachi et al., 2014*; *Earp & Newton, 2012*).

## 30-15 VIFT

The test consisted of 30-s shuttle runs separated by 15-s passive recovery periods. The initial 30-s run started at 8 km/h, increasing by 0.5 km/h every subsequent 45-s stage. The target distances for each 30 s were calculated while considering the increasing effort required for direction changes. An empirical value of 0.7 s was deducted from each 30-s run to account for direction changes, and the final running speed was recorded. Participants completed the shuttle run over a 40-m distance, running between two lines. Pre-recorded beeps controlled their speed, and they adjusted it by reaching 3-m safety zones located at the track's ends and centre. During the 15-s recovery period, participants walked forward to the closest starting line for the next run, depending on where they stopped in the previous stage (either in the middle or at the end of the running area). The test continued until exhaustion, with termination occurring when a participant failed to reach the designated safe zone three consecutive times. The final speed that reached the point of exhaustion was recorded as the VIFT value (*Buchheit et al., 2009*). Additionally, estimated $VO_{2max}$ (ml/kg/min) values were calculated from the VIFT result. Participants received verbal encouragement to perform at their peak throughout the test (*Buchheit & Laursen, 2013*). The $VO_{2max}$ formula is presented below, and the test procedure is illustrated in Fig. 3.

$$VO_{2max} = 28.3 - 2.15 \times Sex(Male = 1, Female = 2) - 0.741(Age) - 0.357(Bodymass) + 0.0586(Age)(VIFT[speed, km/h]) + 1.03(VIFT[speed, km/h].$$

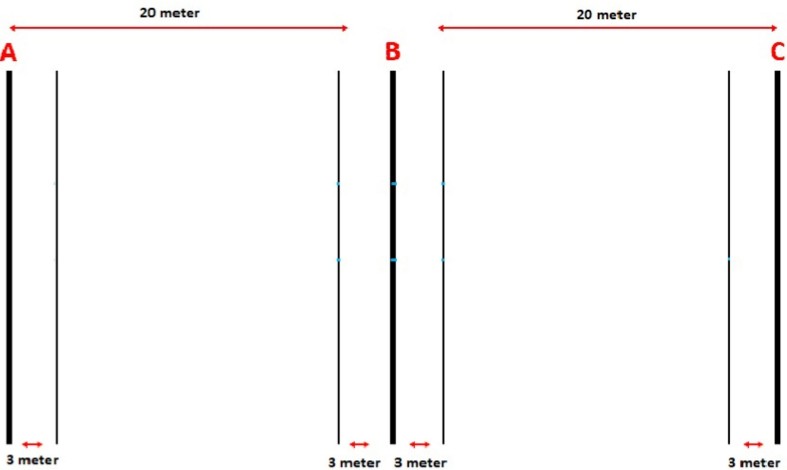

**Figure 3  30-15 IFT test protocol.**

## Statistical analysis

The dataset was primarily examined for erroneous values, outliers, and multicollinearity. To assess the distribution's conformity to a normal curve, skewness and kurtosis were examined. As the calculated coefficients for both metrics fell between −1.5 and +1.5, the data were deemed to follow a normal distribution (*Kim, 2013*). The SPSS 25 software package was used to analyse the data. The demographic characteristics of the athletes were analysed using descriptive statistics (IBM Corp., Armonk, NY, USA). The findings are expressed as the mean ± standard deviation (x ± SD). Due to the verification of normal distribution criteria, parametric statistical methods were applied. A two-way repeated measures ANOVA (2 groups × 2-time points) was utilised to detect differences in the maximum sprint speed (MSS), COD, VIFT and $VO_{2max}$ parameters. The Bonferroni test compared group and time changes for variables with significant group-time interactions. Greenhouse–Geisser corrections were applied to F tests when Mauchly's test of sphericity was violated. Additionally, partial eta squares ($\eta^2 p$) were calculated for the effect size. The effect size obtained from $\eta^2 p$ was grouped as large if ≥0.14, medium if ≥0.06, and small if <0.05 (*Richardson, 2011*).

## RESULTS

When the analysis results in Table 2 were examined, a group-time interaction was observed in all parameters ($p < 0.001$). In the experimental group, MSS significantly increased in the post-test ($F_{(1, 18)} = 21.489$, $p < 0.001$, $\eta^2 p = 0.642$), with a 3.79% increase and a 0.40% increase in the control group. COD time also significantly improved in the experimental group in the post-test ($F_{(1, 18)} = 51.418$, $p < 0.001$, $\eta^2 p = 0.811$), demonstrating a large effect size. There was a 2.21% decrease in the experimental group and a 0.40% increase in the control group. According to the 30-15 IFT results, the post-test values in the experimental group showed a significant increase ($F_{(1, 18)} = 33.330$, $p < 0.001$, $\eta^2 p = 0.692$). While a 5.05% increase in performance was observed in the

**Table 2 The effects of vHITT training on various performance parameters.**

| | Exp (n = 7) | | Con (n = 7) | | Interaction | | |
|---|---|---|---|---|---|---|---|
| | Pre test $\bar{x} \pm$ SS 95% CI | Post test $\bar{x} \pm$ SS 95% CI | Pre test $\bar{x} \pm$ SS 95% CI | Post test $\bar{x} \pm$ SS 95% CI | F | p | $\eta^2$p |
| MSS (km/h) | 25.09 ± 0.61 [24.52–25.66] | 26.04 ± 0.64 [25.45–26.63] | 25.01 ± 0.68 [24.38–25.64] | 25.11 ± 0.56 [24.59–25.63] | 21.489 | <0.001 | 0.642 |
| COD (sec) | 5.42 ± 0.18 [5.25–5.58] | 5.30 ± 0.17 [5.14–5.46] | 5.42 ± 0.08 [5.35–5.49] | 5.43 ± 0.09 [5.34–5.50] | 51.418 | <0.001 | 0.811 |
| VIFT | 18.42 ± 0.60 [17.86–18.99] | 19.35 ± 0.69 [18.71–19.99] | 18.29 ± 0.57 [17.76–18.81] | 18.50 ± 0.58 [17.96–19.03] | 33.330 | <0.001 | 0.735 |
| VO$_{2max}$ | 48.60 ± 1.19 [47.49–49.71] | 50.91 ± 1.66 [49.37–52.45] | 48.42 ± 1.37 [47.14–49.69] | 48.88 ± 1.39 [47.59–50.16] | 25.782 | <0.001 | 0.682 |

Note:
Exp, experimental; Con, control.

experimental group, only a 1.15% increase was observed in the control group. Similarly, VO$_{2max}$ increased by 0.95% in the control group compared to a 4.76% increase in the experimental group (F(1, 18) = 25.782, $p < 0.001$, $\eta^2$p = 0.682) (Fig. 4). In addition, all variables showed a statistically significant effect ($\eta^2$p ≥ 0.14) (Table 2).

## DISCUSSION

This study primarily aimed to evaluate the effects of a four-week vHIIT training program on key athletic performance metrics, including MSS and COD endurance. The research compared the performance of soccer players who underwent vHIIT training twice a week for four weeks, in addition to their regular training, with a CON that continued with only regular soccer training without any additional endurance training. Soccer players participating in the vHIIT training program showed significant improvement in MSS, with a large effect size, indicating enhanced sprint performance. The COD performance also demonstrated a large effect size, highlighting improvements in agility and movement efficiency. Additionally, VIFT performance and VO$_{2max}$ showed substantial physiological improvements. These findings suggest that vHIIT improves anaerobic and aerobic performance in young soccer players.

High-intensity sprint activities hold a vital role in soccer performance (*Faude, Koch & Meyer, 2012*). The average game consists of intensive 10–30-m sprints performed in 2–5-s intervals (*Girard, Mendez-Villanueva & Bishop, 2011*). Advanced energy metabolism and neuromuscular system adaptations become necessary for performing complex high-intensity actions. Research findings demonstrate that the implemented intervention significantly enhanced participants' maximum sprint speed performance compared to those in the control group. Research evidence demonstrates that HIIT exercise effectively enhances sprint performance, including maximum speed achievement (*D'Isanto et al., 2022*; *Kunz et al., 2019*). Performance adaptations remained limited exclusively to the intervention group, while the CON demonstrated no measurable changes, thereby confirming the necessity of specific training solutions for normalising performance.

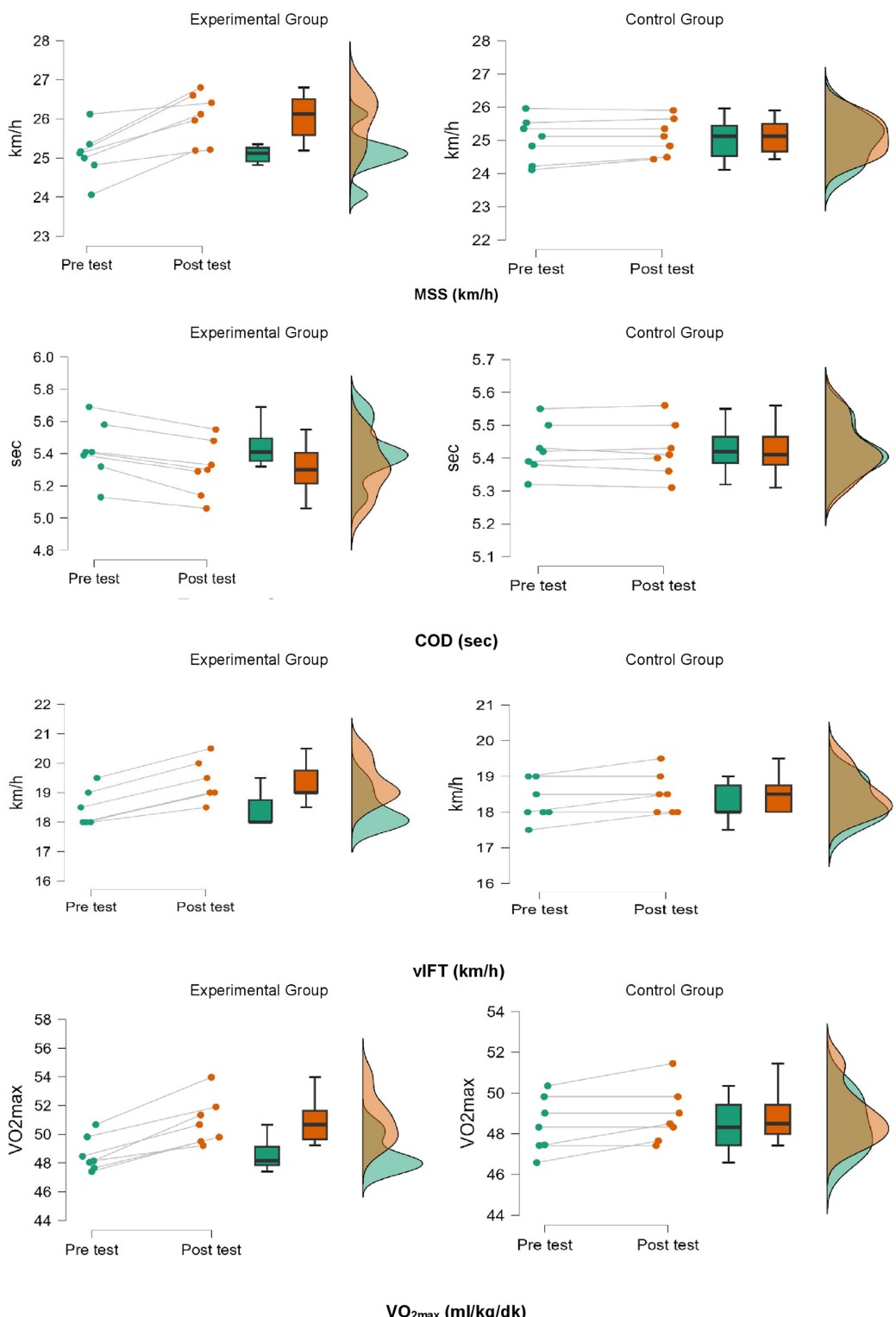

**Figure 4 Comparison of pre-test and post-test parameters of groups.**

Studies have shown that HIIT exercise has positive effects on athletic sprinting abilities (*Engel et al., 2018*; *Engel & Bauer, 2022*). One study concluded that HIIT training improved the speed and anaerobic capacity of athletes (*Jatmiko, Kusnanik & Sidik, 2024*). Performance enhancement in sprints becomes possible through ventilatory high-intensity interval training because it enhances both energy metabolism systems and neuromuscular development (*Arazi et al., 2017*; *Fan et al., 2024*; *Milioni et al., 2024*). HIIT training promotes muscle fiber changes because it enhances the number of type IIb muscle fibers thus accelerating the maximum sprint capabilities of soccer athletes (*Castillo et al., 2021*)—vHIIT training results in improved sprint performance due to structural changes in humans' anaerobic energy managing systems. The improvements to the ATP-PCr system enhance soccer's maximum sprint speed.

Soccer players execute instant COD movements during both offensive and defensive operations. Modern soccer's rapid pace increases the need for players to perform activities that require them to successfully change their mass flow from vertical to horizontal directions based on their game positions. To achieve peak performance, a person must develop their neuromuscular capacity to the highest level (*Morin et al., 2015*; *Öztürk et al., 2023*). According to research findings, the experimental group attained significantly better COD test outcomes than the control group. While research has shown that HIIT training improves sprint and change of direction performance abilities in soccer players (*Michailidis et al., 2022*), another study has revealed that HIIT training shows strong connections with soccer-related COD performances (*Clemente et al., 2021*).

In a study investigating the HIIT of the Shuttle Run on an athlete's COD ability and anaerobic capacity, both locomotor performances were shown to increase (*Jatmiko, Kusnanik & Sidik, 2024*). HIIT exercises work to build anaerobic capacity because this ability directly supports high-intensity direction changes (*Engel & Bauer, 2022*; *Sitzmann, Akrama & Baumann, 2021*). Enhanced anaerobic capacity helps neuromuscular adaptation by strengthening the motor control system (*Twist, Bott & Highton, 2023*). The research indicates that improved motor control systems result in better agility, alongside increased COD speed during complex movement sequences (*Kobal et al., 2017*; *Stølen et al., 2005*). HIIT delivers a stimulus to a range of body positions and equilibrium systems while enhancing the functioning of motor units and motor control abilities (*Abuwarda, Mansy & Megahed, 2024*). Research shows that vHIIT training produces its strongest effects on soccer players' anaerobic capacity but simultaneously leads to neuromuscular improvements that enhance motor control performance. The improved performance functions directly enhance COD performance. Neuromuscular system adaptations explain the performance enhancement observed in COD after vHIIT training.

The participants in the intervention group experienced significant improvements in endurance, whereas the participants in the control group did not demonstrate comparable results. According to *D'Isanto et al. (2022)*, young soccer player participants saw their $VO_{2max}$ performance improve by 7.0% through HIIT training. Research on soccer players has shown that they gain significant benefits by increasing their endurance performance through high-intensity interval training (*Ramadhan, Alim & Ayudi, 2022*). A study investigating the effects of HIIT training on futsal players saw a 22.1% increase in their

$VO_{2max}$ (*Fatemeh, Ramin & Marzieh, 2016*). Research evidence indicates HIIT training generates improved endurance capabilities and increased $VO_{2max}$ metrics for athletic performers (*Arazi et al., 2017*; *Bahtra et al., 2023*; *Clemente et al., 2024*; *Jatmiko, Kusnanik & Sidik, 2024*; *Sepang et al., 2023*). Studies conducted in science prove that high-intensity interval training produces maximal performance results following sessions that exceed 85% of maximum intensity (*Ariningsih, 2021*; *Blagrove, Howatson & Hayes, 2018*).

The research results demonstrate that soccer players benefit from vHIIT workouts at their maximal oxygen utilisation rate, enhancing endurance and maximum oxygen consumption performance. The endurance performance improvements from $VO_{2max}$-based HIIT training occur through multiple physical changes within the body. The exercise regimen known as HIIT facilitates cardiovascular development, characterised by increased stroke volume, along with enhanced oxygen transportation to active muscles (*Hostrup & Bangsbo, 2023*). HIIT training promotes mitochondrial formation, which strengthens muscle oxidative capacity and enhances match performance abilities (*Ariningsih, 2021*; *Batacan et al., 2017*; *Muminović, Pržulj & Jovanović, 2022*). For successful high-intensity soccer movements, players must have adaptations that support aerobic and anaerobic energy systems. Improving endurance performance and $VO_{2max}$ measurement requires vHIIT training as an essential component for athletes involved in soccer.

## Limitations

This study has some limitations and may provide direction for future research. According to the statistically calculated G*Power analysis, 34 participants were required. However, because the study was conducted within a single sports club to minimise external factors that could arise from different coaches, training centres, or training content and to ensure training standardisation, 14 male soccer players completed the study. Future research should expand the athlete sample base to encompass both professional and amateur levels, as this specialised sample provides important findings but does not address performance differences. Research would need to extend beyond small groups to better understand the relationship between the findings and professional soccer teams and athletes of diverse ages. The research design limited its findings to male subjects, which may limit the generalizability of its conclusions to male gender groups. By including female players in additional research studies, scientists can enhance the validity of their research and the relevance of their results to the broader soccer-playing community. Assessing enduring performance development is challenging since the research period was limited to four weeks. This study encounters an important drawback because it cannot conclusively state the long-term advantages. The fact that the total training load (volume and intensity) was not matched between groups should be considered a limitation of the study. $VO_{2max}$ was obtained by formulating VIFT scores. However, this formula may be limited to small groups. Pre- and post-intervention assessments were conducted by the same experienced researcher. Standard protocols were applied to minimise measurement bias. However, the lack of blinding of the assessor regarding group assignment may be considered a

limitation. Additionally, although large effect sizes were observed, the limited sample size warrants cautious interpretation when generalising these findings to broader populations.

## CONCLUSION

Research findings demonstrate that 4-week HIIT training at a high intensity with a speed focus effectively boosts the sprint capability, change of direction, and the endurance performance of soccer players. According to these research findings, soccer coaches and athletic performance trainers can enhance running distance and speed through vHIIT applications. These fundamental athletic qualities are critical for controlling the flow of play in positions that rely on speed and agility. Coaches can enhance soccer player performance in maximum sprint speed, COD, and endurance levels through vHIIT training sessions created according to 30-15 IFT test outcomes. Future research needs to investigate how vHIIT training affects performance at various levels of athletes participating in diverse sports disciplines. The combination of vHIIT training with common strength and conditioning drills used in soccer, including resistance training, plyometric training, small-sided games, and speed, agility, quickness (SAQ) drills, can generate valuable insights about enhanced performance outcomes.

### Recommendations

To comprehensively understand the impact of the vHIIT training program on athletic performance, implementing longer intervention durations may provide more consistent and reliable results. Additionally, future studies could be designed to compare heart rate-based HIIT with speed-based HIIT training, providing insights into the most effective training methodology for improving athletic performance.

To evaluate the effects of vHIIT more objectively, a methodological balance should be established by ensuring equal total training volume and intensity for both groups. This way, the effects of vHIIT can be demonstrated more clearly, and potential overestimated results can be avoided.

Additionally, since physical performance differences are not evident in regional youth players according to their field positions (*Sampaio et al., 2023*), coaches should structure training programs based on individual player performance profiles rather than field positions.

### Funding

Prince Sultan University provided support in covering the Article Processing Charges for this publication, assistance in paying publication fees, and the allocation of research resources. The funders had no role in study design, data collection and analysis, decision to publish, or preparation of the manuscript.

## Grant Disclosures

The following grant information was disclosed by the authors:
Prince Sultan University.

## Competing Interests

The authors declare that they have no competing interests.

## Author Contributions

- Murat Koç conceived and designed the experiments, performed the experiments, prepared figures and/or tables, authored or reviewed drafts of the article, and approved the final draft.
- Niyazi Sıdkı Adıgüzel conceived and designed the experiments, performed the experiments, prepared figures and/or tables, authored or reviewed drafts of the article, and approved the final draft.
- Hakan Engin conceived and designed the experiments, performed the experiments, prepared figures and/or tables, authored or reviewed drafts of the article, and approved the final draft.
- Barışcan Öztürk conceived and designed the experiments, performed the experiments, authored or reviewed drafts of the article, and approved the final draft.
- Umut Canli conceived and designed the experiments, performed the experiments, authored or reviewed drafts of the article, and approved the final draft.
- Aydın Karaçam conceived and designed the experiments, performed the experiments, authored or reviewed drafts of the article, and approved the final draft.
- Bekir Erhan Orhan conceived and designed the experiments, performed the experiments, authored or reviewed drafts of the article, and approved the final draft.
- Pablo Prieto-González conceived and designed the experiments, analyzed the data, prepared figures and/or tables, authored or reviewed drafts of the article, and approved the final draft.
- Peter Bartik conceived and designed the experiments, analyzed the data, authored or reviewed drafts of the article, and approved the final draft.
- Shahad Alghemlas conceived and designed the experiments, analyzed the data, authored or reviewed drafts of the article, and approved the final draft.
- Maria Isip conceived and designed the experiments, analyzed the data, authored or reviewed drafts of the article, and approved the final draft.
- Peter Sagat conceived and designed the experiments, analyzed the data, prepared figures and/or tables, authored or reviewed drafts of the article, and approved the final draft.

## Ethics

The following information was supplied relating to ethical approvals (*i.e.*, approving body and any reference numbers):

Ethics Committee of Bandırma Onyedi Eylül University, Health Sciences Non-interventional Research, 20-12-2024-1042/241.

## Data Availability

Data is available in the Supplemental Files.

## Supplemental Information

Supplemental information for this article can be found online at http://dx.doi.org/10.7717/peerj.20066#supplemental-information.

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
