# Peer review of "Effects of 4-week velocity-based HIIT on athletic performance in youth soccer players"

_PeerJ, doi:10.7717/peerj.20066_

## Round 0.1 · original submission · Major Revisions

Dear Authors,

Thank you for submitting your manuscript.

Please revise the document considering the reviewers´ suggestions.

Thank you.

Best regards.

·

Basic reporting

Abstract
Line 31: The main bit missing here is why? Why are you investigating vHIIT in this population? Does it potentially address a problem? Can you briefly say why at the start of the abstract?
Line 36: How long was the intervention? (says in the title but not in the abstract)
Line 37: should state that these assessments were conducted across experimental and control (for clarity)
Lin 42: no significant differences in the control group – from pre to post, in comparison to the experimental group, or both?
Line 47: I’ll be interested to see how you define locomotor performance. I come from a motor competence background (e.g., performing locomotor skills like running, jumping, hopping), and you’re assessing a mix of fitness and agility, rather than locomotor skill. I’ll keep an eye out for it in your introduction.
Introduction
Line 57: L. Radziminski – is this the right citation formatting?
Line 97: I agree with the justification for using vHIIT, but I’m not sure locomotor performance is best labeled as the outcomes seem to be mostly fitness related. What is the argument for locomotor performance over fitness (or fitness-related description)?

Experimental design

Methods
Line 131: How was random assignment to intervention and control conducted? How many clubs were involved? Was it that 14 footballers from one club provided consent, and they were randomly allocated? How did you account for cross-contamination if that was the case? Did you have prior connections to the club or clubs?
Line 132: Some of this section is more procedural around the design and intervention rather than about the number of participants. So I suggest keeping only information pertinent to participants in this section and either moving or integrating more procedural elements into the procedure section you have below.
Line 133: How was the vHIIT embedded? Was it every session in the 4-5 training sessions per week? Was it a section of the training session, and if so, how long? Because this information appears here, it makes you ask these questions when probably these are explained in the procedure section below, so connecting to the comment above, perhaps move this information to the procedure section.
Line 139: The information from here and onwards addresses most of the questions in my comment about line 131. I suggest moving this information earlier in this section and addressing any outstanding questions.
Line 146-148: Should this be under procedure rather than participants?
Line 150: Here, you say it’s athletic performance, not locomotor performance, which I think is more appropriate.
Line 179: Did the seca stadiamoter measure mass as well?
Line 181: maximal instead of maksimal?

Validity of the findings

Line 241: Here again, you say athletic performance rather than locomotor performance, so I would change it to this, meaning a change in the title and other areas.
Line 242: apologies, I may have missed this, it says twice a week here, was this information put in the procedure? I can’t see it. Also, did this replace two of the 4-5 sessions they had? This should be clearer. Did the control group participate in training sessions that were similar to the non-VHIIT sessions completed by the experimental group?
Lines 278-281: Did this study and SR have a relative HIIT dosage to your study?
Line 296-3097: throughout this section, and other places, it seems that HIIT is really effective, so why don’t footballers receive this type of training more widely? Is there a reason that coaches are not taking up this approach, and is there a way this can be changed? It’s great to show effective improvement, but will this study end up creating more evidence that it works, but not formulating ways for it to be picked up more widely? Or at least identifying reasons it’s not being used and presenting ways to move forward?
Generally, discussion should include a strengths and limitations section, so I suggest adding one.

·

Basic reporting

General comments
This manuscript addresses the effects of a 4-week velocity-based high-intensity interval training (vHIIT) program on key locomotor performance variables in youth soccer players. The topic is timely and relevant, particularly in the context of evidence-based conditioning strategies for young athletes. The use of a randomized controlled design, inclusion of field-relevant performance tests (MSS, COD, vIFT, VO₂max), and alignment with sport-specific demands are commendable. However, there are important concerns regarding sample size justification, methodological clarity (especially regarding dropout and compliance), and the consistency between the introduction, methods, and conclusions. Thus, some sections need some major revisions to improve the methodological issues and the overall results.

Specific comments
Title: The authors should make it clear which target population fits the ‘youth soccer player’ analysed.
Abstract: In the inferences presented, the statistics are missing (lines 40-44). In the conclusions, you should also abbreviate change of direction (COD) (line 45).
Introduction: The rationale of the study is well presented, and the research topic is interesting, and the research question is clear and addresses a gap regarding vHIIT in youth soccer players. Please, provide some normative values found for HIIT training for young soccer players (lines 68-98).
I would recommend expanding the comparison to confounding variables that are often presented in this type of analysis, such as positional differences (https://peerj.com/articles/15609/) or maturational variables (which don't seem to have been analysed). Also, more complex indices such as reactivity indices, or reactive strength ratios (https://doi.org/10.3389/fspor.2024.1282214) (lines 95-108).

Experimental design

Material and Methods:
Lines 116-120: A randomized controlled trial design was adopted. However, the sample size was insufficient according to the authors' own power calculation (n = 34 required, only 14 completed). This seriously undermines the study’s power and generalizability. Authors must either (i) acknowledge this as a limitation or (ii) reframe the design as a pilot study. The randomization process is not described in sufficient detail. How was allocation concealment ensured?
Lines 139-145: Dropouts (n=4) are reported but not analyzed (e.g., intention-to-treat). Reasons for dropout should be presented in detail and discussed.
Lines 166-173: The vHIIT protocol description is sound, but the control group’s training load is not described. Please clarify whether training volume/intensity was matched or recorded.

Validity of the findings

Results: The pre/post testing was comprehensive and covered relevant performance metrics. However, although the results are clearly reported with appropriate effect sizes (η²p), the F statistics and the qualitative magnitude of the effects should also be mentioned.
The data analysis would benefit from confidence intervals and exact p-values where possible. Assumptions of normality and sphericity are discussed, but raw statistical outputs (e.g., Mauchly's test results) are not shown. Consider including them in the supplementary material.
It is not clear whether a blinded assessor performed the tests. If not, this is a potential source of bias.
The validity of the VO₂max estimate derived from the 30-15 IFT formula should be discussed. This is not a direct physiological measurement and may be less reliable in small samples.
The small sample size makes it difficult to draw robust conclusions. This should be clearly stated in the abstract and conclusions.

Discussion
The inferences should not be presented again in the discussion, only key outcomes. Please remove (e.g., partial eta squared = 0.642, p < 0.001) (lines 244-250).
The discussion covers a wide range of related literature and reinforces the training value of vHIIT. The interpretation of results should be more critical. For example, large effect sizes in small samples should be cautiously discussed to avoid overinterpretation. The comparison to HR-based HIIT is relevant but underdeveloped. Expand on the practical advantages and disadvantages of vHIIT in real-world settings.
The authors should better acknowledge the limitations, particularly related to: small sample size; lack of follow-up (no data on retention or long-term effects); absence of training load monitoring in the control group.
Future directions are mentioned but would benefit from concrete proposals (e.g., multicentre trials, use of physiological markers, sex differences). Also, the contextualisation of the results with the tactical and behavioural functions (https://peerj.com/articles/14381/)

Conclusion: The conclusion is generally supported by the data, but should be tempered in light of the small sample size. Phrases like “effectively boosts” could be rephrased to reflect a more cautious tone, such as “may contribute to improvements in…”
References: Revise the style and form according to the instructions given by the salesperson. An upgrade is recommended.

---

## Round 0.2 · Major Revisions

Dear Authors,

Please revise the manuscript considering the suggestions from Reviewer 1.

Thank you.

Best regards.

·

Basic reporting

Thank you for addressing all my suggestions at the review stage. I have some further comments, mostly minor.
Abstract
You did say that you changed locomotor performance to athletic performance however locomotor performance is still used in the abstract.
Line 40: if the vHIIT was provided in addition to the weekly training received in the experimental group, this should be inserted here.
Line 44: “at pre-test” rather than “as pre-test”?
Conclusion: you may have addressed this in the main manuscript however I’ll flag it here first; it could be argued that extra training of any sort may lead to increases in athletic performance, as in, if I do 6 training sessions in comparison to my team mate who does 4, I could be more fit than they are, regardless of the type of training I did. Apologies if you have already addressed this or if the other reviewer commented on this, but was dosage similar between control group and experimental group but the experimental group received vHIIT as part of their dosage? Or were there different dosages of training between control and experimental group? If dosage was different, this will need to be addressed.
Introduction
Line 113: you don’t need to put the Fitton Davies reference there, the Arazi reference is sufficient.

Experimental design

Methods
There still some sections of writing that should really appear in other places, for example:
You have the sample size determination section which is fine but then should have a “Participants” sub-heading for the paragraph under that one. Line 150-153 is part of the design rather than participants so should be moved there.
The last sentence about injury is an eligibility criterion and should stay with the participant section. On that, were there other eligibility/inclusion criteria that should be noted here and any exclusion criteria?
I would then put the experimental design section next, and then the procedures as the measures can come directly after the procedure then.
Line 158: this may reveal my ignorance on this but what does, “Randomisation impartiality was maintained throughout the process” mean?
The procedures section reads a little muddled, with it jumping between the vHIIT procedure and the pre-post test procedure. If the “experimental design” section covers what they did during vHIIT then procedures section should mostly revolve around the order and timing of the pre-test procedures for experimental and control, then I’d say something like, “After the 4-week vHIIT implementation, post test was conducted…” then go on to say how post-test measures were run. You have most if not all elements there, just needs restructuring for full clarity.
Line 192: irregular use of capitalisation for body mass and height measurement
Line 197: Maxsimal or maximal?
Line 224: was this encouragement given at certain times or whenever effort seemed to be dwindling? Essentially, was encouragement standardised across participants?

Validity of the findings

Results
I’d suggest placing the text above the tables and then refer to the tables in the text (like you have done).

Discussion
Line 269: Faude reference mentioned twice, should remove one. This also happens in line 278 to 280 with Engel inserted 4 times.
Line 280: because you have written Jatmiko as part of the sentence, you do not need to put it again in brackets at the end of the sentence. This will apply in all other places this occurs too.
Line 341: controlling game flow is repeated here
Limitations section should be part of the discussion, not after the conclusion, whereby future research directly comes after, limitations-future directions-conclusion
Line 362: yes the unequal training load is a limitation but I think you need to go further and state that this makes the results preliminary effective and research needs to be conducted to test vHIIT where groups have equal dosage to understand exactly if vHIIT is as effective as hypothesised.

·

Basic reporting

Dear Author,

After a careful and detailed response from the reviewers, I recommend accepting the manuscript in its present form. Congratulations.

Experimental design

Nothing to add.

Validity of the findings

Nothing to add.

Additional comments

Nothing to add.

---

## Round 0.3 · Minor Revisions

Please revise the manuscript considering the minor revision recommendation by reviewer 1. Thank you. Best regards.

·

Basic reporting

Abstract
Line 53: …which are fundamental requirements of soccer.

Introduction
Line 88: there’s an “&” when it should be “and” as it is in the main text. You also don’t need to cite it again in brackets at the end of this sentence.
Line 91: think there should be a “.” rather than “-“ before “the training…”
Line 107: “…of the average age (17.4) in 4 weeks of HITT…” does not read quite right. Please revise.
Line 106: 2013 needs to be in brackets
Line 108: you don’t need Faude in brackets here as you mention this study in the actual sentence.
Lines 109-112: you have three sentences in a row starting with “However…” Please revise.

Experimental design

Methods
Line 131: I’d put the duration of intervention before the vHIIT here
Line 142: 34 was needed as minimum; I have a couple of questions just on this: 1) did you then try to over recruit (above 34) to account for any attrition from pre to post-test? (or due to injury or sickness), 2) How many players did you try to recruit? Was this over 34? As it reads currently, it sounds like 20 players were recruited and allocated into experimental or control, then 6 players were withdrawn, so how many ended in the experimental and how many in the control? I cannot see this allocation described anywhere, apologies if I missed this. Just need a little more clarity on this.
Line 204: what do you mean by “age at sports”?

Validity of the findings

Results
Line 264: (Table 2) should be in the sentence, not separated.
This section only states the improvement at post-test for the experimental group, there is no reference to group x time effects or any comparison to the control group. Please review how to present the F statistic. LAERD is a very good statistical resource that shows how to present different statistical results.

Discussion
Line 271: again here, like in the results, it only refers to the experimental group, which implies a time only effect, but was there a group x time effect? If so, you need to compare with the control group.
Line 324: (Ramadhan et al., 2022) should be in the sentence, not separated.
You state that the limitations had been moved but they are still after the conclusion. Limitations and future directions should come before the conclusion.
Line 357: 14 is also a lot lower than the minimum of 34 as calculated by your sample size calculation.

---

## Round 0.4 · accepted · Accept

Dear Authors,

Thank you, and congratulations on your work during the review process, addressing all of the reviewers' comments.

Best regards.